# Randomised Controlled Feasibility Study of the MyHealthAvatar-Diabetes Smartphone App for Reducing Prolonged Sitting Time in Type 2 Diabetes Mellitus

**DOI:** 10.3390/ijerph17124414

**Published:** 2020-06-19

**Authors:** Daniel P. Bailey, Lucie H. Mugridge, Feng Dong, Xu Zhang, Angel M. Chater

**Affiliations:** 1Division of Sport, Health and Exercise Sciences, Department of Life Sciences, Brunel University London, Kingston Lane, Uxbridge UB8 3PH, UK; 2Centre for Human Performance, Exercise and Rehabilitation, Brunel University London, Kingston Lane, Uxbridge UB8 3PH, UK; 3Institute for Sport and Physical Activity Research, School of Sport Science and Physical Activity, University of Bedfordshire, Polhill Avenue, Bedford MK41 9EA, UK; luciemugridge@googlemail.com (L.H.M.); Angel.chater@beds.ac.uk (A.M.C.); 4Department of Computer and Information Sciences, University of Strathclyde, Glasgow G1 1XH, UK; feng.dong@strath.ac.uk; 5Institute for Research in Applicable Computing, University of Bedfordshire, Luton LU1 3JU, UK; xuzhang2015@gmail.com

**Keywords:** sedentary behaviour, glucose, health apps, behaviour change, theory of planned behaviour, health coaching

## Abstract

This study evaluated the feasibility and acceptability of a self-regulation smartphone app for reducing prolonged sitting in people with Type 2 diabetes mellitus (T2DM). This was a two-arm, randomised, controlled feasibility trial. The intervention group used the MyHealthAvatar-Diabetes smartphone app for 8 weeks. The app uses a number of behaviour change techniques aimed at reducing and breaking up sitting time. Eligibility, recruitment, retention, and completion rates for the outcomes (sitting, standing, stepping, and health-related measures) assessed trial feasibility. Interviews with participants explored intervention acceptability. Participants with T2DM were randomised to the control (*n* = 10) and intervention groups (*n* = 10). Recruitment and retention rates were 71% and 90%, respectively. The remaining participants provided 100% of data for the study measures. The MyHealthAvatar-Diabetes app was viewed as acceptable for reducing and breaking up sitting time. There were preliminary improvements in the number of breaks in sitting per day, body fat %, glucose tolerance, attitude, intention, planning, wellbeing, and positive and negative affect in favour of the intervention group. In conclusion, the findings indicate that it would be feasible to deliver and evaluate the efficacy of the MyHealthAvatar-Diabetes app for breaking up sitting time and improving health outcomes in a full trial.

## 1. Introduction

The global prevalence of diabetes is estimated to be 415 million [1]. Type 2 diabetes mellitus (T2DM) accounts for 87–91% of all cases of diabetes and increases the risk of cardiovascular disease (CVD), premature mortality, poor psychological wellbeing, and reduced quality of life [2,3]. Effective interventions to manage glycaemia are thus important for reducing the onset of these T2DM-related health outcomes [4].

Engaging in high amounts of sedentary behaviour is detrimentally associated with the risk of diabetes, CVD, mortality, and poor mental health, independent of physical activity levels [5,6,7]. In people with T2DM, increased sedentary time is associated with higher levels of CVD risk markers, including hyperglycaemia, insulin resistance, and low high-density lipoprotein cholesterol [8]. Conversely, an increased number of breaks in sedentary time per day is beneficially associated with glucose tolerance and triglyceride levels [9]. Experimental studies have also consistently reported attenuations in postprandial glycaemia in response to regularly breaking up sitting time with light or moderate-intensity physical activity over a single day [10,11,12,13,14]. People with T2DM have been found to engage in 9.5 h/day of device-measured sedentary time [15]. Physical activity guidelines for this population thus suggest reductions in total daily sitting time and regular interruptions to prolonged sitting [16]. However, there is a paucity of research that has tested interventions for reducing sitting time in people with T2DM or to understand the determinants of sitting time in this population using theoretical frameworks [17], such as the Theory of Planned Behaviour [18]. Web-based interventions using such theories have, however, shown promise in this population to enhance physical activity levels through intervening with attitudes, subjective norms, perceived behavioural control, and intentions [19].

It is suggested that smartphone apps have the potential to be used as low-cost health behaviour change interventions [20]. In the UK, 78% of people used a smartphone in 2018 [21], and there are currently 325,000 mobile health apps available [22]. There have been a limited number of studies evaluating smartphone app interventions for reducing sitting time [23,24]. These apps include features to enable monitoring of sitting time, goal-setting, prompts to interrupt sitting, and feedback on behaviour. App-based interventions have resulted in reductions in total sitting time and an increase in the number of breaks in sitting per day after one week in the general adult population [23,24]. However, these apps were only tested over a short time period (one week). In a small sample of people with T2DM (*n* = 7), daily sedentary time responses to an app that prompted them to break up their sitting time were varied, from decreases by −13% to increases by 11% [25]. The number of breaks in sedentary time also unfavourably decreased by 16 breaks per day. These interventions have often not been explicitly linked to theoretical models or modes of effective delivery, such as motivational interviewing [26] and health coaching [27]. There has also been limited focus on utilising behaviour change techniques (BCTs) within these apps [28] to target reductions in total daily and prolonged sitting time [23]. The aim of this study was, therefore, to evaluate the feasibility and acceptability of a self-regulation smartphone app with targeted BCTs for reducing prolonged sitting in people with T2DM.

## 2. Materials and Methods

### 2.1. Study Design and Overview

This was a two-arm, randomised, controlled feasibility trial reported in line with the Consolidated Standards of Reporting Trials (CONSORT) guidelines [29]; see Appendix A. Ethical approval was granted by the Cambridge South National Health Services (NHS) Research Ethics Committee (reference 17/EE/0070) and was registered on the ISRCTN clinical trials registry (number ISRCTN11167977). The trial was carried out in accordance with the Declaration of Helsinki. All testing procedures took place at the University of Bedfordshire Sport and Exercise Science Laboratories. After baseline data collection, participants were randomised in blocks of four with a ratio of 1:1 by the research team to either the control or intervention group, using an online tool (www.randomization.com). Participants in the control group received usual care. As this was a feasibility study, the sample size was pragmatic and not based on a power calculation.

### 2.2. Participants

Participants were recruited from local GP surgeries, a Diabetes UK support group, and social media. Inclusion criteria were: males and females aged 18–65 years; self-reported T2DM diagnosed within the past four years (early stage) and in the first stage of drug treatment, or using a diet and exercise management strategy only; able to read and speak English; able to stand and walk unassisted; and previous experience using a smartphone. Exclusion criteria included: BMI ≥ 40 kg/m^2^; cardiovascular disease, kidney disease or hypertension; severe mental health concerns; and pregnancy. Participants were email or telephone-screened for eligibility, and provided written informed consent prior to any testing procedures.

### 2.3. Intervention

Participants randomised to the intervention group were asked to use the MyHealthAvatar-Diabetes smartphone app for 8 weeks. MyHealthAvatar is a platform developed by the European Commission-funded research project “MyHealthAvatar: Demonstration of a 4D Digital Avatar Representation for Access to Long-Term Health Status Information” led by the University of Bedfordshire to provide digital representation of patients’ health statuses with the aim of creating a “digital avatar” of the user [30]. The infrastructure enables the collection, access, and sustainability of health and lifestyle information and is informed by a survey of user needs [31]. The MyHealthAvatar-Diabetes app is compatible with Android devices and was developed for the management of health and lifestyle behaviour in patients with diabetes. This app was designed in collaboration between the authors, and draws on a number of BCTs [28] to encourage reductions in sitting time, as described below. It specifically uses the BCTs: goal-setting, action planning, review behaviour goals, discrepancy between current behaviour and the goal, review outcome of goal, feedback on behaviour, self-monitoring of behaviour, feedback on outcomes of behaviour, information about health consequences, monitoring of emotional consequences, and prompts/cues. Participants were provided with an Android smartphone to use if they did not have access to one. The app comprises a range of suites accessible to the user to encourage reductions in sitting time and improved management of their disease, as described below.

#### 2.3.1. MyHealthAvatar-Diabetes App

##### Sitting Behaviour Suite

Sitting time and breaks in sitting per day are monitored in real-time, using a combination of the smartphone’s built-in accelerometers and gyroscope sensor. This allows monitoring of the angle change rate for movement in three planes (see Appendix A). Sitting and standing postures are detected using the Y-axis. The general accuracy of mobile sensors for monitoring body positions when the phone is carried in the pocket is over 90% [32]. If the phone was being used for a call or any other purpose (meaning the screen was on), this was classified as sitting time if the phone sensors detected no other movement. Users are able to set self-generated personalised goals and monitor their progress towards these goals with regard to their total sitting time and breaks in sitting per day. The data is displayed to the user (see Figure 1) and progress towards each goal is shown as a bar filling and the corresponding percentage towards that goal. For total daily sitting, 100% would mean that the user has achieved the maximum amount of time sitting that day relevant to their goal, while 100% for breaks would mean that they have achieved their daily goal of breaks in sitting per day. An alert to prompt participants to take a break from sitting was automatically set to occur every 30 min during the day and read “It is time to take a break from sitting”. Users were able to achieve a maximum of one break within each 30 min period (e.g., they could achieve a maximum of 20 breaks over a 10 h period). A break was defined as engaging in ≥2 min of continuous ambulation and/or standing [10,11,12], which is displayed to the user. All intervention participants were asked to carry the smartphone in their front trouser pocket throughout the study to maximise the phone’s accuracy for detecting sitting time. The app has a calendar feature where the user can view their data for previous days. The calendar feature works across all of the suites described here.

##### Physical Activity Suite

The number of steps and time spent in physical activity are monitored in real-time using the phone’s in-built accelerometer and location tracker. Goals can be set for the number of daily steps, and the progress toward the daily goal is displayed as a percentage.

##### Body Weight, Glucose, and Blood Pressure Suites

There are separate suites where users can enter and monitor their body weight, glucose, and blood pressure levels (see Appendix A). If participants wished to use these suites, they were responsible for obtaining their own readings using devices that they may have available to them at home, for example. Values could be entered daily and a graph could be displayed, showing their values over time. Goals could be set for body weight, and progress towards their goal would be displayed in percentage terms. For the body weight suite, participants could also enter their height so that their BMI would be calculated and displayed. The app displays whether the user’s BMI is within the underweight, normal weight, overweight, or obese range.

##### Medication Alerts, Mood, and Diabetes Information Suites

Users can set medication alerts to occur at relevant times throughout the day based on their personal medication regime. An alert will appear on the screen accompanied by an alarm at the times specified. The mood suite enables the user to enter their current mood at any time during the day on a five-point scale of “very sad” to “very happy”. The average mood of the entries for that day would then be displayed graphically to the user (see Appendix A). Participants are also able to complete a weekly mood questionnaire via the Positive and Negative Affect Scale [33], which would then display their positive and negative affect scores. The information section of this suite encouraged the user to “try to improve your mood by being more active and sitting less”. There is also a suite for users to access information on diabetes, where there are links to recent diabetes-related NHS news articles.

#### 2.3.2. Text Message Support

In addition to the MyHealthAvatar-Diabetes app, participants received two text messages per week, sent on Monday and Friday from the research team to encourage reductions in sitting. Text message content was based on motivational interviewing [26] and the Goal, Reality, Opportunity, and Will (GROW) model of health coaching [27] (see Appendix A) and used a combined communication approach to deliver the BCTs to enhance intrinsic motivation, as previously recommended [34,35,36].

### 2.4. Data Collection

Data collection took place at baseline and 8 weeks (end of the intervention period). Participants were asked to fast for ≥10 h and not consume alcohol or caffeine for 24 h prior to the morning of each data collection session. They were required not to engage in any exercise for 48 h prior to data collection, and to travel to the laboratories by car to minimise their activity levels during that morning. Upon arrival, participants had a fasting blood sample taken, and then completed an oral glucose tolerance test (OGTT). They then had other anthropometric and cardiometabolic health measures taken, and completed questionnaires as described below.

### 2.5. Primary Outcomes (Feasibility and Acceptability)

Eligibility, recruitment, and retention rates were recorded to assess trial feasibility in addition to completion rates for the outcomes that would be planned in a full trial. Furthermore, acceptability of the intervention was assessed using semi-structured interviews with each intervention participant to explore the experience of using the intervention and barriers and facilitators to their engagement with the MyHealthAvatar-Diabetes app. 

### 2.6. Secondary Outcomes

#### 2.6.1. Sitting, Standing, and Stepping

An activPAL activity monitor (PAL Technologies Ltd., Glasgow, Scotland) was attached to each participant’s right thigh using a medical-grade dressing (Hypafix, BSN medical, Hull, UK) to measure sitting, standing, and stepping outcomes. The activPAL was wrapped in a nitrile sleeve so that it was waterproof. Participants were asked to wear the device continuously for eight consecutive days at baseline and during the last week of the intervention. They were provided with a diary to record sleep and wake times and any periods where the device was removed. The activPAL provides a valid and reliable measure of sitting, standing, stepping, and postural transitions [37,38]. 

The activPAL data was processed in STATA (StataCorp LLC, College Station, TX, USA) using an automated algorithm [39]. A valid day of wear was considered a wear time of >10 h, at least 500 steps, and no more than 95% of data in one activity category (i.e., sitting, standing, or stepping). For quality control, each valid day was visually compared with diary notes to confirm waking wear time. The specific outcomes for this study derived from the activPAL included daily sitting, standing, stepping, light-intensity stepping, and moderate-to-vigorous stepping, each expressed as a percentage of daily waking wear time to account for any differences in wear time between baseline and follow-up. Light and moderate-to-vigorous stepping intensities were calculated by the activPAL’s internal algorithm that is based on step rate [40]. The number of breaks in sitting, number of prolonged sitting bouts, and total steps per day were also derived. The average across all valid days was calculated for each of these outcomes.

#### 2.6.2. Anthropometry and Cardiometabolic Health

Height was measured using a stadiometer. Body weight and body fat % were measured using the TANITA BC-418 Segmental Body Composition Analyzer (Tanita Corp., Tokyo, Japan) which uses bio-electrical impedance analysis. The waist circumference was measured at the level of the umbilicus using an adjustable tape measure. Resting blood pressure was measured using the Omron M5-I automatic blood pressure monitor (Omron Matsusaka Co Ltd., Matsusaka, Japan). A finger prick blood sample was collected for measurement of fasting glucose, which was followed by an OGTT, whereby each participant consumed 75 g of glucose (100% dextrose monohydrate powder; Thornton & Ross Ltd., UK) mixed with 300 mL of water. A further finger prick blood sample was taken 2 h after consuming the glucose load for measurement of 2-h glucose. A volume of 30 μL was collected for each blood sample, and capillary blood glucose concentrations were measured immediately using the YSI 2300 STAT plus glucose and lactate analyzer (YSI Inc., Yellow Springs, OH, USA).

#### 2.6.3. Determinants of Sitting Behaviour, Mood, and Wellbeing

Determinants of sitting behaviour were measured based on the Theory of Planned Behaviour [41] using standardised wording formats [42], with the behavioural outcome termed “avoid long periods of sitting” and the duration being over a one-week period, anchored on a seven-point likert scale. This included four items on attitude (e.g., “Avoiding long periods of sitting would be harmful/beneficial”), four items on subjective norms (e.g., “Most people who are important to me think that I should avoid long periods of sitting over the next week”), three items on perceived behavioural control (e.g., “Whether I avoid long periods of sitting over the next week is entirely up to me”) and three items on intentions (e.g., “I intend to avoid long periods of sitting over the next week”). Current mood was assessed using the Positive and Negative Affect Scale [33] and wellbeing using the Office for National Statistics National Wellbeing Measurement [43] and the Warwick Edinburgh Mental Wellbeing Scale [44].

### 2.7. Data Analysis

Data analysis included calculating eligibility (participants eligible/participants assessed for eligibility × 100), recruitment (participants randomised/number of eligible participants screened × 100), and retention (participants completing the intervention/participants enrolled × 100) rates, in addition to completion rates for the data collection measures (participants providing full outcome data/participants completing the study × 100). Interviews were audio-recorded and transcribed verbatim. Thematic analysis was used to analyse the interviews [45], which permitted in-depth exploration of participant perceptions of the intervention. In accordance with recommendations for pilot and feasibility studies, significance testing was not conducted, as it is inappropriate to place undue significance on hypothesis testing when no formal power calculations have been performed [46]. Furthermore, there was likely to be an imbalance in pre-randomisation covariates with small sample sizes, which would need to be adjusted for in the analysis and which could compromise statistical power [46]. Therefore, this study calculated Cohen’s d effect sizes to explore preliminary trends in the data (between-group differences). Effect sizes of 0.2, 0.5, and 0.8 indicated a small, medium, and large effect, respectively. Data are reported as mean (SD). Descriptive variables were generated using SPSS V.22 (IBM, Armonk, NY, USA).

## 3. Results

### 3.1. Feasibility

Figure 2 shows participant progression throughout the study. Participants were recruited from January 2017 until June 2017 on a rolling basis, with the last follow-up assessment taking place in August 2017. Of the 50 individuals screened prior to the eligibility assessment, 26% did not respond to further contact. Of the 37 participants assessed for eligibility, 24% (*n* = 9) did not meet the inclusion criteria and 22% (*n* = 8) declined to take part (eligibility rate = 76%). Nine males and 11 females were enrolled into the study and randomised to the control or intervention group (recruitment rate = 71%). Table 1 shows the descriptive characteristics of the participants.

One participant withdrew from each arm, meaning that nine participants in each group completed the study (retention rate = 90%). The 18 participants that completed the study provided 100% of data for each of the study measures.

### 3.2. Participants’ Views on Using the MyHealthAvatar-Diabetes App

There were four themes that were identified from the interviews with regard to participant views of the acceptability and feasibility of the MyHealthAvatar-Diabetes app. These were (1) “Prompting behaviour change”, (2) “Sense of achievement”, (3) “Technical issues”, and (4) “Environmental barriers”. 

#### 3.2.1. Theme 1—Prompting Behaviour Change

Participants highlighted that the app prompted them to change their behaviour. This was facilitated by the alert raising their awareness and reminding them to take a break from sitting, and led to self-realisation of how much time they were spending in prolonged sitting.

“The most useful thing with the app is that it would vibrate every 30 min, which meant I didn’t have to look at it, I just had it in my pocket and when I felt it vibrate, I would just automatically stand up” (male, aged 44).

“It made me realise how often I do sit for exceptionally long periods of time, especially during the workday when you get engrossed in something and you don’t leave, so having the app that did alert you was useful” (female, aged 54).

#### 3.2.2. Theme 2—Sense of Achievement

Participants reported how the app helped to increase their motivation to reach sitting time and step goals, which led to a sense of achievement. The goal-setting function of the app was identified as being particularly helpful in this respect, in addition to being able to self-monitor progress towards goals in real-time.

“Goals were probably one of the best bits in the app; it showed you how many periods you had stood up for” (male, aged 44).

“It certainly made me think about how I sit all the time and my frame of mind has changed, I’m more positive now and I’m out walking with my sister” (female, aged 63).

#### 3.2.3. Theme 3—Technical Issues

Interest and adherence to using the app may have been affected by technical complications. Participants explained that they would prefer the break reminder alerts to be reactive to the time they had spent sitting, as opposed to being set to occur every 30 min. Some participants also felt that the number of steps being recorded was inaccurate when comparing it to other apps and wearable devices they had access to, such as a Fitbit or Apple iPhone step-counter.

“Every half hour it bleeps at you rather than recording what you are doing and reacting to it. If you want people to get up every half hour, it needs to tell you when you have sat down” (male, aged 65).

“I’m not sure it was consistent in how it recorded, there was some days where I felt I had done more walking and I felt it was inconsistent” (female, aged 53).

#### 3.2.4. Theme 4—Environmental Barriers

Environmental barriers to using the app were commonly described, such as ensuring the phone was charged, remembering to have the phone with you and in your pocket at all times, and having to wear the phone in a trouser pocket to enable accurate detection of sitting time.

“Remembering to have it in my pocket and also it limits you to what you can wear, I wear a lot of these jeggings and they don’t have the pockets [in order to carry the phone]” (female, aged 63).

“Just the fact that it’s a mobile phone, it’s not easy to always have it on you. I didn’t always have clothes where I had a pocket and I didn’t really feel it was that accurate because of that” (female, aged 60).

### 3.3. Changes in Sitting, Standing, and Stepping

The analysis of between-group differences from baseline to follow-up found a large effect for the change in the number of breaks in sitting per day in favour of the intervention group (see Table 2). Both the control and intervention groups decreased their percentage of time stepping, but the reduction was smaller in the intervention group (medium effect). There was also a medium effect for the change in light stepping time in favour of the intervention group.

### 3.4. Changes in Anthropometric and Cardiometabolic Outcomes

There was a medium effect for the change in body fat % and 2-h blood glucose in favour of the intervention group (see Table 3). The effect sizes for the between-group differences were trivial or small for the remaining anthropometric and cardiometabolic outcomes.

### 3.5. Changes in Determinants of Sitting Behaviour, Mood, and Wellbeing

For the psychological variables (Appendix A), in relation to the Theory of Planned Behaviour, there was a medium effect for changes in attitude in favour of the intervention. Perceived behavioural control increased in both groups, but more so in the control group. Subjective norms reduced in the intervention group, relevant to the controls. Self-efficacy increased in favour of the control group, reducing in the intervention group, and planning increased in favour of the intervention group. All elements of wellbeing (general wellbeing, positive affect, and items related to life satisfaction, worthwhileness, and happiness) changed in favour of the intervention (small to large effect sizes), while negative affect reduced in favour of the intervention, and anxiety made no change in the intervention group but reduced in the controls. 

## 4. Discussion

This study demonstrated the feasibility of implementing and evaluating the MyHealthAvatar-Diabetes smartphone app for reducing prolonged sitting in people with early-stage T2DM. It was deemed feasible to deliver and evaluate the intervention, evidenced by the low dropout rate during the study and high compliance with providing outcome measure data. Valid activPAL data was collected from all of the participants who completed the study, indicating efficacy of collecting primary outcome data (sitting time) in a fully powered RCT. Those allocated to the intervention found the app to be an acceptable intervention tool that they cited was particularly useful for raising awareness of their sitting levels, prompting changes in sitting behaviour and providing a sense of achievement. Prompting behaviour change was facilitated by the alerts to take a break from sitting and the real-time self-monitoring of sitting goals. The use of prompts and self-monitoring of sitting time via computer and smartphone apps has been documented in previous interventions that have effectively reduced sitting time and increased the number of breaks from sitting in adults and office workers [23,24,47]. The sense of achievement that resulted from using the MyHealthAvatar-Diabetes app appeared to be due to the ability to set sitting goals and monitor progress towards these goals in real-time. There have been a number of studies published in other population groups in which goal-setting and self-monitoring have been used as part of an effective multicomponent intervention to reduce sitting [48,49,50]. This study demonstrates the acceptability of delivering specific BCTs via a smartphone app for reducing prolonged sitting in people with T2DM.

This study demonstrated the preliminary efficacy of the MyHealthAvatar-Diabetes app for increasing the number of breaks from sitting per day. A smartphone app (stAPP) aimed at reducing sitting behaviour that included similar functions to MyHealthAvatar-Diabetes (e.g., real-time self-monitoring, prompts, goal-setting) significantly increased the number of breaks per day (+5.7 breaks/day) in adults of the general population after one week [23]. Similar to the present study, the stAPP app did not lead to any changes in total daily sitting time. However, neither of these studies were powered to detect changes in total sitting time. Another smartphone app (B-MOBILE) that comprised of sedentary behaviour goal-setting, prompting, and feedback significantly reduced total daily sitting time after one week in overweight/obese adults [24]. This may have been due to the B-MOBILE app prompting participants to break up their sitting time with 3, 6, or 12 min of walking every 30, 60, or 120 min, respectively. Furthermore, the app had a greater focus on goals and self-monitoring of activity breaks than the MyHealthAvatar-Diabetes and stAPP apps, which may have thus resulted in a greater displacement of sitting time. The MyHealthAvatar-Diabetes app thus appears to have promise for increasing the number of breaks from sitting, but may need refinement to encourage reductions in total sitting time in a full trial.

The preliminary effects analysis suggested that the MyHealthAvatar-Diabetes app may increase the percentage of time spent in light-intensity stepping and total stepping. The B-MOBILE app significantly increased percentages of time spent in light physical activity, and also increased time spent in moderate-to-vigorous physical activity [24]. This is in contrast to the stAPP app that did not cause any changes in physical activity outcomes [23], which may have been because the stAPP app did not provide any information on steps (monitored in real-time with MyHealthAvatar-Diabetes and B-MOBILE apps). Functions that enable monitoring of steps thus appear to be important for increasing light or moderate physical activity in apps designed to reduce sitting time.

To the authors knowledge, this is the first study to evaluate the potential effects of a sitting reduction smartphone app on health outcomes. Participants that used the MyHealthAvatar-Diabetes app for 8 weeks had a trend of reduced body fat % and 2-h glucose. This could have been due to an increase in the number of breaks from sitting or increases in light stepping achieved by the intervention group. The capability to set medication reminders and goals in relation to body weight and glucose within the app may also help with improving these health markers. Further research is needed to evaluate the effects of the MyHealthAvatar-Diabetes app on sitting behaviour, physical activity, and health outcomes in a full trial to corroborate the preliminary effects seen in this study.

This study was novel in that it used measurements from the Theory of Planned Behaviour to investigate the determinants of sitting behaviour. As the data is underpowered, definitive conclusions cannot be made. However, overall intention increased in favour of the intervention, as did attitude, suggesting that the intervention may have a positive effect on motivation. Perceived behavioural control also increased, suggesting that the participants’ belief in their ability to break up their sitting was improved. However, the subjective norms of others wanting them to break up their sitting was negatively impacted upon in those who received the intervention, suggesting perceptions of social influences may need to be targeted in future interventions. All aspects of wellbeing showed a positive change in favour of the intervention, which shows promise for a fully powered intervention. 

### Strengths and Limitations

The main strength of this study was the evaluation of an app that was designed using specific BCTs to encourage reductions in prolonged sitting. This study also established the acceptability and feasibility of evaluating the MyHealthAvatar-Diabetes app, and the findings can thus be used to inform a full trial. The limitations of this study include the small sample size, which was based on pragmatism. This may thus limit the generalisability of the findings. However, the feasibility of evaluating the MyHealthAvatar-Diabetes app in a full trial can be approached with confidence, given the high compliance and low dropout rates observed. Although the MyHealthAvatar-Diabetes app was deemed to be acceptable by the participants, there were a number of issues that could have limited user engagement. One issue that was often described was the app’s alert to prompt the user to take a break from sitting being set to automatically occur every 30 min during the day. The participants indicated that they would have preferred the alerts to be reactive, so that they occurred after the user had been sitting for 30 min. This modification has since been made to the app for use in a full trial. Another barrier was the necessity to carry the phone in a trouser pocket for accurate detection of sitting or standing postures. This was particularly problematic for women who often did not wear clothing that would allow this. These issues should be considered when developing sitting-based smartphone apps to maximise user engagement. As discussed previously, it should also be considered how the MyHealthAvatar-Diabetes app could be adapted in a future trial to encourage reductions in total daily sitting time. In this study, we did not stipulate that participants should not use other apps or devices that they already had access to. This was because we wanted to evaluate an intervention that was in addition to participants’ usual behaviour and healthcare. The current intervention could have encouraged increased engagement with such apps or devices, which could have affected their behaviour. Another potential limitation is that the intervention was evaluated only in people with early-stage T2DM. This limits the generalisability to individuals who have had the disease for a longer period of time or who are at a more advanced stage. However, our aim was to evaluate if the intervention could be an effective tool to help manage T2DM at an early stage. It would also be of value to establish the validity of the MyHealthAvatar-Diabetes app for measuring sitting behaviour to further support its use as a public health intervention tool and a method for evaluating responses to sedentary behaviour interventions. Lastly, some of the participants were provided with an Android smartphone if they did not have access to one already e.g., because they owned an iPhone. This could have affected their engagement with the MyHealthAvatar-Diabetes app, although this was not mentioned as a problem during the interviews.

## 5. Conclusions

In conclusion, this study suggests that it would be feasible to evaluate the efficacy of the MyHealthAvatar-Diabetes app for breaking up prolonged sitting in people with T2DM in a fully powered RCT. The intervention was found to be acceptable, although there are a number of changes that could be made to the app to increase participant engagement. The MyHealthAvatar-Diabetes app appeared to have the potential to improve measures of health and wellbeing, further emphasising the need to evaluate the effects in a full trial. 

## Figures and Tables

**Figure 1 ijerph-17-04414-f001:**
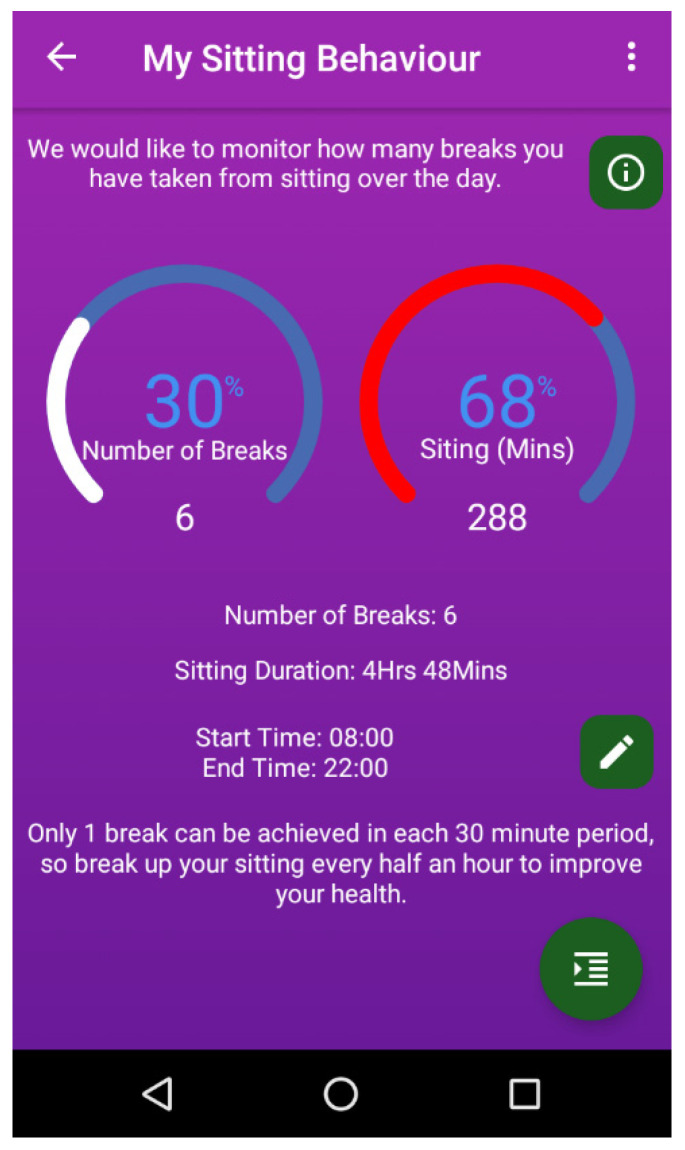
MyHealthAvatar-Diabetes app sitting suite.

**Figure 2 ijerph-17-04414-f002:**
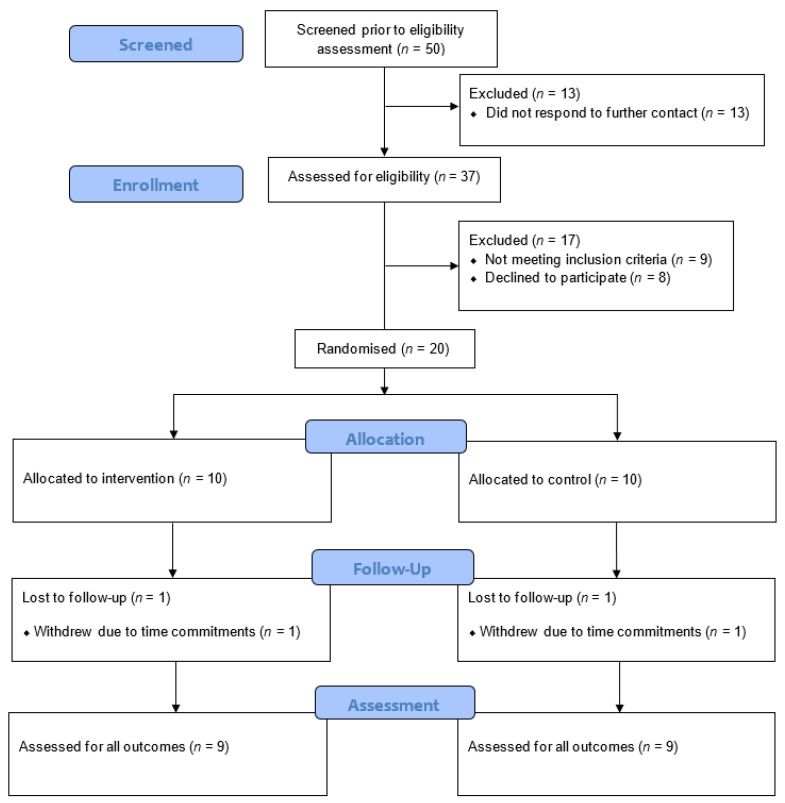
CONSORT diagram of participant progression through the study [29].

**Table 1 ijerph-17-04414-t001:** Participant characteristics. Data presented as mean (SD).

Characteristic	Control Group	Intervention Group
Age	55 (6)	57 (7)
Male (*n*)	6	3
Female (*n*)	3	6
Body mass index (kg/m^2^)	29.9 (4.7)	31.1 (6.4)
Body fat %	31.3 (7.8)	36.9 (9.9)
Waist circumference (cm)	107.2 (11.6)	104.6 (14.9)
Resting systolic blood pressure (mmHg)	134 (18)	136 (17)
Resting diastolic blood pressure (mmHg)	83 (11)	84 (9)
Fasting glucose (mmol/L)	6.57 (1.77)	5.76 (0.99)
2-h glucose (mmol/L)	11.64 (3.06)	10.23 (1.86)

**Table 2 ijerph-17-04414-t002:** Within- and between-group differences from baseline to follow-up for sitting, standing, and stepping outcomes. Data presented as mean (SD).

Variable	Control Baseline	Control Follow-Up	Within-Group Differences	Intervention Baseline	Intervention Follow-Up	Within-Group Differences	Between-Group Differences	Cohen’s d
Waking wear time (min)	959.5 (61.7)	948.2 (70.2)	−11.3 (87.1)	936.3 (54.7)	914.6 (55.5)	−21.7 (28.2)	−10.4 (92.6)	0.16
% Sitting	56.8 (13.9)	53.2 (13.8)	−0.7 (6.2)	68.2 (8.9)	66.1 (9.6)	−2.1 (7.3)	−1.4 (10.1)	0.21
% Standing	31.2 (11.4)	33.4 (12.5)	2.1 (7.4)	21.6 (5.6)	24.0 (6.6)	2.5 (5.2)	0.4 (9.8)	0.06
% Stepping	12.0 (3.8)	10.5 (3.5)	−1.5 (3.1)	10.3 (4.2)	9.9 (3.5)	−0.4 (2.6)	1.1 (4.4)	0.38
% Light stepping	4.9 (1.5)	4.5 (1.6)	−0.4 (1.4)	3.5 (1.1)	3.9 (1.2)	0.3 (0.8)	0.7 (1.7)	0.61
% MVPA stepping	7.0 (2.4)	6.0 (2.2)	−1.1 (2.0)	6.7 (3.3)	6.0 (2.5)	−0.7 (2.0)	−0.4 (3.2)	0.20
Breaks in sitting per day	53.8 (19.2)	49.3 (15.6)	−4.4 (15.7)	47.2 (9.9)	51.6 (14.8)	4.3 (6.5)	8.8 (16.9)	0.72
Prolonged sitting bouts per day	5.0 (1.6)	4.7 (2.1)	−0.3 (1.6)	5.7 (1.1)	5.8 (1.4)	0.1 (1.3)	0.4 (2.5)	0.27
Steps per day	8742.7 (2827.4)	7492.2 (2796.1)	−1250.4 (2467.5)	7772.7 (3689.8)	7109.3 (2986.7)	−663.3 (2318.1)	587.1 (3330.0)	0.25

**Table 3 ijerph-17-04414-t003:** Within- and between-group differences from baseline to follow-up for anthropometric and cardiometabolic outcomes. Data presented as mean (SD).

Variable	Control Baseline	Control Follow-up	Within-Group Differences	Intervention Baseline	Intervention Follow-Up	Within-Group Differences	Between-Group Differences	Cohen’s d
Weight (kg)	90.2 (19.9)	90.5 (19.2)	0.2 (1.7)	89.6 (20.3)	90.0 (21.7)	0.4 (2.8)	0.1 (3.0)	0.09
Body fat %	31.3 (7.8)	31.5 (7.8)	0.3 (1.5)	36.9 (9.9)	35.9 (9.2)	−0.9 (2.6)	−1.2 (3.2)	0.57
Body mass index (kg/m^2^)	29.9 (4.7)	29.9 (4.6)	0.1 (0.6)	31.1 (6.4)	31.2 (6.9)	0.1 (1.1)	0.0 (1.1)	0.00
Waist circumference (cm)	107.2 (11.6)	107.9 (11.6)	0.7 (3.6)	104.6 (14.9)	104.7 (14.1)	0.1 (2.1)	−0.6 (3.7)	0.20
Heart rate (bpm)	66.7 (10.9)	65.9 (10.0)	−0.8 (5.4)	63.0 (13.8)	61.6 (9.8)	−1.4 (7.6)	−0.7 (12.2)	0.09
Systolic blood pressure (mmHg)	134.0 (18.1)	135.6 (20.3)	1.6 (12.4)	136.3 (17.2)	138.2 (20.7)	1.9 (13.3)	0.3 (23.2)	0.02
Diastolic blood pressure (mmHg)	83.3 (10.7)	84.3 (13.2)	1.0 (7.5)	83.8 (9.5)	82.4 (9.8)	−1.3 (9.5)	−2.3 (13.5)	0.27
Fasting blood glucose (mmol/L)	6.55 (1.76)	6.73 (2.66)	0.19 (1.30)	5.75 (1.01)	5.66 (1.20)	−0.09 (0.66)	−0.28 (1.30)	0.27
2-h blood glucose (mmol/L)	11.47 (2.96)	11.22 (3.74)	−0.25 (1.78)	10.46 (1.67)	9.56 (1.20)	−0.90 (1.26)	−0.65 (2.69)	0.42

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
