# Peer review of "Randomised Controlled Feasibility Study of the MyHealthAvatar-Diabetes Smartphone App for Reducing Prolonged Sitting Time in Type 2 Diabetes Mellitus"

_ijerph, 2020, doi:10.3390/ijerph17124414_

Round 1
Reviewer 1 Report
This is a well-written paper of a small study evaluating feasibility and acceptability with positive results as the outcome. In general, all that is required is to fix up some formatting issues with titles and Table 3 as well as breaking up the Discussion section into both a Discussion section and a Limitations section. Here are the specific suggestions for edits.
52-3 As “however” is not the beginning of a new phrase, there should be a comma rather than semi-colon before it—“have, however,”.
73 Change “reported in line the” to “reported in line with the”.
105-6 Line 87 mentions that participants had to have previous experience using a smartphone, yet here it is stated that “participants were provided with an Android smartphone to use if they did not have access to one.” If it were important that participants have previous experience with using the smartphone and some of the participants were given one then it can be assumed that the participants given a smartphone did not have experience with the phone given to them. As a result, if it matters that they have experience, this was a limitation of the study.
125 Did it matter in which trouser pocket the smartphone was kept (front or back)? If so, this should be mentioned.
139 Change “can also enter” to “could also enter”
180 Change “provided a diary” to “provided with a diary”.
213 Change “week’.” to week’)”
249 This appears to be a title for the section to follow. It should be written in the style of other titles and have a blank line to follow.
273 Title should be similar to other titles. Change “3.1. Technical issues” to “Theme 3 - Technical issues”.
293 Change “3.2” to “3.1”.
301 As some of the entries that are too long for the cell have the (SD) listed on the same line, it would perhaps be better if all (SD) were listed on the line below the entry to make the tables easier to read.
303 Change “3.3” to “3.2”.
306 As some of the entries that are too long for the cell have the (SD) listed on the same line, it would perhaps be better if all (SD) were listed on the line below the entry to make the tables easier to read. The size of the first column to the left should be increased so that “baseline” is all on one line as in Table 2. The size of the fourth column from the left should be increased so that “Interven” and “baseline” are both on one line as in Table 2. The size of the fourth column from the right should be increased so that “Interven” is all on one line as in Table 2. The size of the second last column to the right should be increased so that “Between” is all on one line as in Table 2.
307-8 For Table 2, the title appears above the chart. As such, for consistence, the title of Table 3 should appear above the chart, not below.
309 Change “3.4” to “3.3”.
338-347 This represents part of the limitations of the study. It should be in a separate section titled “Limitations”.
353-360 This statement should be added to the new Limitations section.
360-2 The statement in the Discussion should be “The MyHealthAvatar-Diabetes app thus appears to have promise for increasing the number of breaks.” The new Limitations section should have this statement, “The MyHealthAvatar-Diabetes app may need refinement to encourage reductions in total sitting time in a full trial.”
380 Change “While” to “As”.
384-386 Add this statement to the new Limitations section: “However, the subjective norms of others wanting them to break up their sitting was negatively impacted upon in those who received the intervention, suggesting perceptions of social influences may need to be targeted in future interventions.”
392-400 Add these statement to a new Limitations section.
516 This reference was last accessed 10 October 2016. It should be reaccessed in 2020 to ensure the link remains operational.
Author Response
Response to reviewer 1
This is a well-written paper of a small study evaluating feasibility and acceptability with positive results as the outcome. In general, all that is required is to fix up some formatting issues with titles and Table 3 as well as breaking up the Discussion section into both a Discussion section and a Limitations section. Here are the specific suggestions for edits.
Response: We thank the reviewer for their positive comments regarding our manuscript. Please find amendments described below in line with your suggestions.
52-3 As “however” is not the beginning of a new phrase, there should be a comma rather than semi-colon before it—“have, however,”.
Response: This has been corrected on line 52.
73 Change “reported in line the” to “reported in line with the”.
Response: This has been corrected on line 73.
105-6 Line 87 mentions that participants had to have previous experience using a smartphone, yet here it is stated that “participants were provided with an Android smartphone to use if they did not have access to one.” If it were important that participants have previous experience with using the smartphone and some of the participants were given one then it can be assumed that the participants given a smartphone did not have experience with the phone given to them. As a result, if it matters that they have experience, this was a limitation of the study.
Response: it is correct that al participants were required to have experience using a smartphone. This could be any type of smartphone, such as Apple or Android. The app used in this study was only available on Android devices. It is possible that participants without prior experience using an Android device could have had different engagement with the app, although this was not mentioned in the interviews when assessing intervention acceptability. This has been discussed on lines 423-426.
125 Did it matter in which trouser pocket the smartphone was kept (front or back)? If so, this should be mentioned.
Response: Yes it needed to be carried in the front pocket, which is now stated on line 127.
139 Change “can also enter” to “could also enter”
Response: This has been corrected on line 141.
180 Change “provided a diary” to “provided with a diary”.
Response: This has been corrected on line 182.
213 Change “week’.” to week’)”
Response: This has been corrected on line 215.
249 This appears to be a title for the section to follow. It should be written in the style of other titles and have a blank line to follow.
Response: The sub-heading as been changed to italics. We request the editorial office to make the necessary change to the line spacing as per the other formatting changes that were made during the editing process.
273 Title should be similar to other titles. Change “3.1. Technical issues” to “Theme 3 - Technical issues”.
Response: This has been corrected.
293 Change “3.2” to “3.1”.
Response: A previous subheading has now been changed to 3.1, so the current 3.2 is now correct.
301 As some of the entries that are too long for the cell have the (SD) listed on the same line, it would perhaps be better if all (SD) were listed on the line below the entry to make the tables easier to read.
Response: The table has now been edited so that the SD is on a separate line and so that the data is more easily viewed.
303 Change “3.3” to “3.2”.
Response: A previous subheading has now been changed to 3.1, so the current 3.3 is now correct.
306 As some of the entries that are too long for the cell have the (SD) listed on the same line, it would perhaps be better if all (SD) were listed on the line below the entry to make the tables easier to read. The size of the first column to the left should be increased so that “baseline” is all on one line as in Table 2. The size of the fourth column from the left should be increased so that “Interven” and “baseline” are both on one line as in Table 2. The size of the fourth column from the right should be increased so that “Interven” is all on one line as in Table 2. The size of the second last column to the right should be increased so that “Between” is all on one line as in Table 2.
Response: The SDs have been move onto separate lines. We request the editorial office to adjust the column widths as requested by the reviewer due to these changes being applied in the editing process. We also request the editorial office to move text in the results as appropriate so that the table is displayed on a single page.
307-8 For Table 2, the title appears above the chart. As such, for consistence, the title of Table 3 should appear above the chart, not below.
Response: This has been corrected.
309 Change “3.4” to “3.3”.
Response: A previous subheading has now been changed to 3.1, so the current 3.4 is now correct.
338-347 This represents part of the limitations of the study. It should be in a separate section titled “Limitations”.
Response: The section discussing limitations of the app identified by the users has now been moved to the limitations section. A new section titled “Strengths and Limitations” has now been added to the manuscript.
353-360 This statement should be added to the new Limitations section.
Response: The authors feel that this discussion (now on lines 357-367) should remain in the discussion section as it is attempting to interpret the findings in the context of other relative studies. We feel the discussion would become disjointed is this was removed and added to the limitations. Furthermore, the interpretation within this section are not limitations, per se, they are proposed explanations for differences in findings between studies. However, we have now recapped in the limitations section the need to consider “how the MyHealthAvatar-Diabetes could be adapted in a future trial to encourage reductions in total daily sitting time” (line 411-413).
360-2 The statement in the Discussion should be “The MyHealthAvatar-Diabetes app thus appears to have promise for increasing the number of breaks.” The new Limitations section should have this statement, “The MyHealthAvatar-Diabetes app may need refinement to encourage reductions in total sitting time in a full trial.”
Response: This has now been added on line 411-413 in the limitations section.
380 Change “While” to “As”.
Response: This has been corrected on line 385.
384-386 Add this statement to the new Limitations section: “However, the subjective norms of others wanting them to break up their sitting was negatively impacted upon in those who received the intervention, suggesting perceptions of social influences may need to be targeted in future interventions.”
Response: As per one of the responses above, the authors do not feel this is a limitation of the study design. Instead, it offers an interpretation of the findings (albeit negative) in the relevant section of the discussion and highlights important avenues for future research. We feel this discussion of the psychological outcomes would be disjointed should we move this to the limitations section.
392-400 Add these statement to a new Limitations section.
Response: A new “Strengths and limitations” section has been created for this section.
516 This reference was last accessed 10 October 2016. It should be reaccessed in 2020 to ensure the link remains operational.
Response: This link has now been re-accessed and is operational – a new access date has been added (line 542).
Reviewer 2 Report
Randomized controlled feasibility study of the MyHealthAvatar-Diabetes smartphone app for reducing prolonged sitting time in T2D patients. The study as a pilot randomized controlled feasibility study assessed eligibility, recruitment, retention and completion of the outcomes of using smartphone app to reduce prolonged sitting time for people living with type 2 diabetes. Authors particularly focused on the feasibility of measuring behavioral change through using smartphone application. Retention and completion rates were high enough to conclude the valid feasibility of the study. However there are some major concerns in methods.
Primary concerns are accuracy and reliability of the data collection. Since participants already addressed this point in the interview, when participants use a smartphone, it should be convinced that the moment breaking prolonged sitting time is accurately measured. Even though a phone is carried in a trouser pocket, a participant can pick up a call at the same time of standing. Or when a phone moves out of a trouser pocket, the phone can read this behavior as ‘move’. This mechanism could misread the behavior. The study should address how participants keep the phone in the pocket consistently in a day and how to prevent misreading the behavior of using a phone (i.e., texting, picking up a call or doing other activity on the phone).
Next concern is whether feedback-based intervention affects the behavior change. Feedback-based smartphone use reported much better outcomes than the use of application or wearable devices only. This intervention could be controversial whether the behavioral change was resulted from the use of application only or an interaction with healthcare professional or investigators. It is well known that participants are more likely to be proactive when receiving a feedback (Reference: Larsen, R. T., Wagner, V., Keller, C., Juhl, C. B., Langberg, H., & Christensen, J. (2019). Feedback from physical activity monitors to enhance amount of physical activity in adults—a protocol for a systematic review and meta-analysis. Systematic reviews, 8(1), 53.; Patel MS, Asch DA, Volpp KG. Wearable Devices as Facilitators, Not Drivers, of Health Behavior Change. JAMA. 2015;313(5):459–460. doi:10.1001/jama.2014.14781). In addition, I wonder that participants were using other form of wearable devices simultaneously? The interview addressed that participants compared the app to the other form of applications. If so, there could be bias because their behavior changes could be contaminated due to the other triggers. It needs to clarify that participants used only this device.
Third question is setting a goal. How did a participant set a goal? Due to different health status and physical function, a goal may need to be set differently. Is it a customized goal or standardized goal?
Fourth, body composition differences between control and intervention group could show substantial effect size after the study. According to table 1, body fat percentage is higher in intervention group whereas waist circumference is lower in the same group. It could be derived from different gender distribution between control and intervention group. Female typically have lower waist circumference as well as high body fat percentage of male. This difference could show moderate effect size in body fat % despite lower differences in behavioral changes. The study needs to match similar characteristics in terms of body composition so that more accurate effect size would be generated.
Therefore, I would recommend major revision.
Author Response
Response to reviewer 2
Randomized controlled feasibility study of the MyHealthAvatar-Diabetes smartphone app for reducing prolonged sitting time in T2D patients. The study as a pilot randomized controlled feasibility study assessed eligibility, recruitment, retention and completion of the outcomes of using smartphone app to reduce prolonged sitting time for people living with type 2 diabetes. Authors particularly focused on the feasibility of measuring behavioral change through using smartphone application. Retention and completion rates were high enough to conclude the valid feasibility of the study. However there are some major concerns in methods.
Response: We thank the reviewer for their comment regarding our manuscript. We have addressed the concerns in the responses below.
Primary concerns are accuracy and reliability of the data collection. Since participants already addressed this point in the interview, when participants use a smartphone, it should be convinced that the moment breaking prolonged sitting time is accurately measured. Even though a phone is carried in a trouser pocket, a participant can pick up a call at the same time of standing. Or when a phone moves out of a trouser pocket, the phone can read this behavior as ‘move’. This mechanism could misread the behavior. The study should address how participants keep the phone in the pocket consistently in a day and how to prevent misreading the behavior of using a phone (i.e., texting, picking up a call or doing other activity on the phone).
Response: If the phone was being used for a call or any other purpose (meaning the screen was on), this was classified as sitting time if the phone sensors detected no other movement at that time (added on lines 114-116). This therefore minimised any potential misclassification of sitting time as standing or movement due to phone use. We appreciate that this may not give 100% accuracy; this is a limitation of all research using smartphones and wrist-worn wearable devices.
Next concern is whether feedback-based intervention affects the behavior change. Feedback-based smartphone use reported much better outcomes than the use of application or wearable devices only. This intervention could be controversial whether the behavioral change was resulted from the use of application only or an interaction with healthcare professional or investigators. It is well known that participants are more likely to be proactive when receiving a feedback (Reference: Larsen, R. T., Wagner, V., Keller, C., Juhl, C. B., Langberg, H., & Christensen, J. (2019). Feedback from physical activity monitors to enhance amount of physical activity in adults—a protocol for a systematic review and meta-analysis. Systematic reviews, 8(1), 53.; Patel MS, Asch DA, Volpp KG. Wearable Devices as Facilitators, Not Drivers, of Health Behavior Change. JAMA. 2015;313(5):459–460. doi:10.1001/jama.2014.14781). In addition, I wonder that participants were using other form of wearable devices simultaneously? The interview addressed that participants compared the app to the other form of applications. If so, there could be bias because their behavior changes could be contaminated due to the other triggers. It needs to clarify that participants used only this device.
Response: We thank the reviewer for raising this point. To clarify, there was no feedback provided by the research team or healthcare professionals regarding sitting behaviour as part of this intervention. Feedback on sitting behaviour was provided to the participants in real-time from the MyHealthAvatar-Diabetes app. Indeed, it was anticipated that this would be one behaviour change technique through which reductions in prolonged sitting would be achieved. This was alongside the behaviour change techniques stated on line 102-105, which were provided through the app: goal setting, action planning, review behaviour goals, discrepancy between current behaviour and goal, review outcome goal, self-monitoring of behaviour, information about health consequences, monitoring of emotional consequences, and prompts/cues.
In this study, we did not stipulate that participants should not use other apps or devices that they already had access to. This was because we wanted to evaluate an intervention that was in addition to usual behaviour and usual healthcare. Indeed, it is highly unlikely any intervention would be able to stop participants from using apps that are built into smartphones at the point of manufacturing e.g. Apple iPhone step counter. This has been discussed as a limitation on lines 413-417.
Third question is setting a goal. How did a participant set a goal? Due to different health status and physical function, a goal may need to be set differently. Is it a customized goal or standardized goal?
Response: The goals set were personal for each participant. This has been clarified on line 116.
Fourth, body composition differences between control and intervention group could show substantial effect size after the study. According to table 1, body fat percentage is higher in intervention group whereas waist circumference is lower in the same group. It could be derived from different gender distribution between control and intervention group. Female typically have lower waist circumference as well as high body fat percentage of male. This difference could show moderate effect size in body fat % despite lower differences in behavioral changes. The study needs to match similar characteristics in terms of body composition so that more accurate effect size would be generated.
Response: We thank the reviewer for identifying this and agree that this could be the case. It must be re-emphasised that this is a feasibility study and changes in the health measures are only preliminary estimates. It is acknowledged that there is likely to be an imbalance in pre-randomisation covariates with small sample sizes (see lines 229-231 and reference Lancaster et al. 2004). However, it is not appropriate to conduct significance testing (including adjusting for covariates) in feasibility trials (see lines 226-228). In a full trial, we agree that this would be required.
Reviewer 3 Report
This research is an interesting approach that attempts to verify the efficacy of an app that reduces sitting time in people with type 2 diabetes. I feel that the experimental design, intervention methods, and application development are excellent, but as the author also states, the sample size is extremely small, and it is extremely disappointing that the sitting behaviors are the secondary outcomes. That is the reason for rejecting this paper.
Other reviewed comments are shown below.
1. In 125 lines, please show some percentage of the total that you could properly evaluate the sitting behavior by putting smartphone in their pants properly.
2. The correlation coefficient of sitting time that can be evaluated with ActivePAL and smartphone should be provided. Furthermore, it should be verified whether the sitting behavior of the smartphone can be properly evaluated based on ActivePAL. By rewriting from this perspective, another paper may be created.
3. In Table 2 & 3, I would like to see the significance level.
Author Response
Response to reviewer 3
This research is an interesting approach that attempts to verify the efficacy of an app that reduces sitting time in people with type 2 diabetes. I feel that the experimental design, intervention methods, and application development are excellent, but as the author also states, the sample size is extremely small, and it is extremely disappointing that the sitting behaviors are the secondary outcomes. That is the reason for rejecting this paper.
Response: We thank the reviewer for their comments and acknowledging the rigorous nature of our study. As stated on line 81, the sample size was pragmatic and not based on a power calculation due to it being a feasibility study. A sample size of n=12 per group has been suggested for feasibility and pilot studies (Julious et al. 2005; doi: 10.1002/pst.185); our study had slightly less participants (n=10 in each group). Despite the relatively small sample size, we were able to answer the study aims around the feasibility and acceptability of a self-regulation smartphone app with targeted BCTs for reducing prolonged sitting in people with T2DM. Indeed, we demonstrated high retention rates and data measure completion rates in this study, in addition to trends in the dating suggesting preliminary efficacy of the app for reducing prolonged sitting and improving some measures of health. Furthermore, data saturation was reached for the interviews that were conducted, again indicating no further participants were required to determine the acceptability of the intervention.
Lastly, the study by Pellegrini et al (2015) referred to in our manuscript used a sample size of n=9, which was sufficient to explore the acceptability of a phone app to break up sitting time. However, they failed to include a control group which we feel is important for the evaluating the acceptability of participant being randomised to the control or intervention groups; this was appropriately explored in our feasibility study.
Regarding the sitting behaviours being secondary outcomes, it is necessary to assess the feasibility of delivering and evaluating a complex intervention prior to assessing effectiveness of an intervention on the intended behaviour outcome to ensure that there is not a waste of resources in a fully powered trial (please see Medical Research Council guidelines on developing and evaluating complex interventions: (https://mrc.ukri.org/documents/pdf/complex-interventions-guidance/)
- In 125 lines, please show some percentage of the total that you could properly evaluate the sitting behavior by putting smartphone in their pants properly.
Response: Please see line 113 that states “The general accuracy of mobile sensors for monitoring body position when the phone is carried in the pocket is over 90% [32]”.
- The correlation coefficient of sitting time that can be evaluated with ActivePAL and smartphone should be provided. Furthermore, it should be verified whether the sitting behavior of the smartphone can be properly evaluated based on ActivePAL. By rewriting from this perspective, another paper may be created.
Response: We thank the reviewer for this suggestion. Due to the study design, it was not possible to validate the MyHealthAvatar-Diabetes app against the activPAL device as this would need to take place following strict protocols under observation from the research team. We have plans to validate the MyHealthAvatar-Diabetes app against the activPAL device in the future. This is explained on lines 421-423.
- In Table 2 & 3, I would like to see the significance level.
Response: In line with recommendations for best practice when conducting pilot and feasibility studies, significance testing was not conducted (Lancaster, G.A., S. Dodd, and P.R. Williamson, Design and analysis of pilot studies: recommendations for good practice. J Eval Clin Pract, 2004. 10(2): p. 307-12). As described on lines 226-228, significance testing was not conducted as it is inappropriate to place undue significance on hypothesis testing when no formal power calculations have been performed. Furthermore, there is likely to be imbalance in pre-randomisation covariates with the relative small sample sizes in feasibility studies and this would need adjusting for in the analysis, which would compromise statistical power. Therefore, this study calculated Cohen’s d effect sizes to explore preliminary trends in the data.
Round 2
Reviewer 2 Report
The revised manuscript has addressed major concerns with additional elaboration in limitation section. There is nothing else to revise it. Congratulations.
Reviewer 3 Report
Although it was confirmed that the revised paper is trying to improve it further, I do not think that the academic value will be significantly improved. My decision remains unpublishable as before.